# A Diamond-Based Dose-per-Pulse X-ray Detector for Radiation Therapy

**DOI:** 10.3390/ma14185203

**Published:** 2021-09-10

**Authors:** Sara Pettinato, Marco Girolami, Riccardo Olivieri, Antonella Stravato, Cristina Caruso, Stefano Salvatori

**Affiliations:** 1Engineering Faculty, Niccolò Cusano University, Via don Carlo Gnocchi 3, 00166 Rome, Italy; sara.pettinato@unicusano.it; 2Istituto di Struttura della Materia, Consiglio Nazionale delle Ricerche (ISM-CNR), Strada Provinciale 35D 9, Montelibretti, 00010 Rome, Italy; 3Azienda Ospedaliera “San Giovanni—Addolorata”, 00184 Rome, Italy; rolivieri@hsangiovanni.roma.it (R.O.); astravato@hsangiovanni.it (A.S.); ccaruso@hsangiovanni.it (C.C.)

**Keywords:** CVD diamond, gated integrator, LINAC, pulsed-mode charge measurements, X-ray detectors

## Abstract

One of the goals of modern dynamic radiotherapy treatments is to deliver high-dose values in the shortest irradiation time possible. In such a context, fast X-ray detectors and reliable front-end readout electronics for beam diagnostics are crucial to meet the necessary quality assurance requirements of care plans. This work describes a diamond-based detection system able to acquire and process the dose delivered by every single pulse sourced by a linear accelerator (LINAC) generating 6-MV X-ray beams. The proposed system is able to measure the intensity of X-ray pulses in a limited integration period around each pulse, thus reducing the inaccuracy induced by unnecessarily long acquisition times. Detector sensitivity under 6-MV X-photons in the 0.1–10 Gy dose range was measured to be 302.2 nC/Gy at a bias voltage of 10 V. Pulse-by-pulse measurements returned a charge-per-pulse value of 84.68 pC, in excellent agreement with the value estimated (but not directly measured) with a commercial electrometer operating in a continuous integration mode. Significantly, by intrinsically holding the acquired signal, the proposed system enables signal processing even in the millisecond period between two consecutive pulses, thus allowing for effective real-time dose-per-pulse monitoring.

## 1. Introduction

In radiation therapy (RT), accurate measurements of delivered doses are crucial both for defining effective treatment plans and for meeting the stringent quality assurance (QA) requirements for patient management and safety. In external beam RT, linear accelerators (LINACs) are used to generate either photon or electron beams that consist of pulses with a duration of few microseconds. In a LINAC, short electron packets are accelerated either to be directly emitted or to impinge on a heavy metal target (e.g., Tungsten) to produce X-ray photons. Pulse repetition rates are typically in the 60–1000 Hz range. Resulting signals are then periodic with a low duty cycle. Modern RT techniques, such as IMRT (intensity-modulated radiation therapy) and VMAT (volumetric-modulated arc therapy) [1,2,3,4], are oriented towards the delivery of large dose gradients in the shortest time possible. IMRT and VMAT are highly conformal techniques that allow for the irradiation of complex and irregularly shaped tumors close to delicate organs that could be damaged if the absorbed dose exceeds a certain tolerance level; specifically, the dose is released by LINAC in a very short time interval, and is appropriately modulated to be at its maximum on the tumor mass and at its minimum at the level of organs at risk [5,6,7]. This ensures an optimal efficacy of treatment, as well as a lower incidence of side effects.

Currently, as a gold standard, a dose is typically measured by ionization chambers, with a collection time in the order of 10^−5^–10^−4^ s [8], coupled to electrometers with relatively long integration times (0.1–10 s [9]). As such, pulse-by-pulse measurements at high repetition rates are not permitted. For this purpose, the demand for fast X-ray detectors coupled to adequate front-end readout electronics, enabling a real-time dose-per-pulse monitoring, is continuously increasing. Aimed at contributing to satisfying such demand, we introduce, in this work, a complete measurement system based on an ultra-fast diamond detector coupled to customized front-end readout electronics designed for pulse-by-pulse signal processing to be specifically used in the development of optimized RT plans for the treatment of solid tumors.

Chemical vapor deposition (CVD) represents the most widely used technique for the production of synthetic diamonds, allowing for the growth of high-quality poly- and mono-crystalline samples. In the case of micro-wave plasma-assisted CVD (MWCVD), which is the technique used for the diamond samples tested in this work, both growth rate and control of purity can be optimized; in “electronic-grade” and “optical-grade” samples, the concentrations of nitrogen and boron impurities can be reduced down to values lower than 5 and 0.5 ppb, respectively, i.e., about three orders of magnitude lower than those reported for the best (namely, type IIa) natural diamond [10]. These features, along with tissue equivalence, high radiation hardness (up to 10 MGy) and high cohesion energy (≈43 eV), make CVD diamond an elective material for the realization of high-performance dosimeters [11,12,13,14] and detectors for X-rays [15,16,17], UV [18,19], charge particles [20,21,22,23] and neutrons [24]. For instance, diamond detectors with long-term durability have already been successfully tested in high-radiation environments, such as those reproduced in ATLAS, a general-purpose particle physics experiment performed at the Large Hadron Collider (LHC) at CERN [25]. Most interestingly, diamond detectors show a response time in the nanosecond range [19,26], making diamonds a unique solid-state material for pulse-by-pulse dosimetry in modern RT. However, if the goal is to measure the dose delivered by every single X-ray pulse, the use of an ultra-fast detector is a necessary but not sufficient condition, as the detector must be coupled to adequate electronics for photogenerated signal processing. In this sense, electrometers are not ideal instruments, as they operate over the total time of irradiation and thus measure a relatively high number of pulses. Some commercial electrometers ensure integration times as low as 0.2 ms, but at the expense of a lower resolution [27], which does not allow for the accurate pulse-by-pulse monitoring of X-ray beams.

The literature on pulse-by-pulse detection systems is scarce, and mainly focused on scintillator- and fiber-optics-based dosimetry [28,29]. As for diamond-based solutions, Velthuis et al. [30] reported on a prototypal system for single-pulse measurements; however, this exhibited a response time of several hundreds of μs and required relatively complex front-end analogue electronics. Conversely, the system we propose here is based on a solid-state CVD-diamond dosimeter coupled to compact and versatile gated-integrator electronics, ensuring response times of a few tens of μs and thus enabling pulse-by-pulse measurements at high repetition rates, as well as inherently allowing for the sample-and-hold operation of charges collected per single pulse.

## 2. Materials and Methods

### 2.1. Pulsed X-ray Source

To explain the rationale behind our system, it is worth describing the source used for generating X-ray pulses, which is a Clinac iX system (Varian Medical Systems) installed at the RT-Department of “San Giovanni-Addolorata” Hospital (Rome, Italy). Figure 1 shows an example of the pulsed signals generated by 6-MV electron packets at different dose rates (DRs).

Red peaks indicate a 360 Hz synchronization signal (sync) made available from the Linear Accelerator (LINAC) console (Varian Medical System, Palo Alto, CA, USA). On the top, the figure shows the signal generated by packets of electrons impinging on the tungsten target, also available from the LINAC console. Signals were acquired by means of a DSO-X-3024A digital oscilloscope (Keysight Technologies, Santa Rosa, CA, USA).

The measurement system we propose here works in sync with the sync signal, i.e., measurements are performed by integrating the charge collected by the diamond detector in a time interval around each X-ray pulse (thus strongly reducing the inaccuracy induced by external noise), whereas processing is carried out in the following hold period. In this sense, the system is able to complete signal acquisition and processing before the arrival of a new pulse, enabling real-time dose-per-pulse monitoring.

### 2.2. Diamond Detector

A 4 × 4 × 0.5 mm^3^ optical-grade single-crystal CVD diamond sample (Element Six), with [N] < 5 ppb and [B] < 0.5 ppb, was used for the fabrication of the detector with a typical metal–semiconductor–metal structure; circular contacts (3.2 mm diameter) were formed on the top and bottom surfaces of the diamond sample by sputter deposition through a shadow mask of a 300 nm-thick Ag film. Contact pads were connected to a 3.5 m-long triaxal cable and the whole detector was then encapsulated into a PMMA cylinder (9 mm diameter). For the tests under 6-MV X-rays, the detector was placed into a Plexiglas^®^ phantom and positioned at the LINAC isocentre. As shown in Figure 2, X-ray response was preliminarily evaluated by measuring the collected charge by means of a Keithley 6517A electrometer (Keithley Instruments & Products, Cleveland, OH, USA). Bias voltage was set to 10 V. Dose was varied in the range 0.1–10 Gy (typically used in radiotherapy). Dose rate was set to 3 Gy/min. Best fit of experimental data returned a linear dependence of charge with dose, with a slope equal to (302.2 ± 0.1) nC/Gy.

### 2.3. Gated Integration Readout Electronics

By taking into account the frequency of X-ray pulses at 3 Gy/min, from the data in Figure 2, a mean charge value <*Q_est_*> ~84.60 pC was subsequently estimated for the charge generated by the single pulse. One of the aims of the work is the real-time direct measurement of collected charge, avoiding an offline estimation. For this purpose, we employed a gated-integrator method, implemented by the electronic system described elsewhere [31,32] and briefly recalled in the following (see Figure 3). The front-end electronics consists of a high-precision-switched integrator transimpedance amplifier IVC102 (Texas Instruments, Dallas, TX, USA). The embedded timer of an LPC845 microcontroller (NXP Semiconductors, Eindhoven, NL, UE) was used to generate digital control signals (S1 and S2) for the IVC102. The timer was programmed to set the start and the end of the integration around a single X-ray pulse; by considering a timer clock frequency of 30 MHz, S1 and S2 signals can be synchronized with the LINAC sync signal within only ±33 ns. A fast comparator (Texas Instruments, Dallas, TX, USA) was used to generate a synchronization signal compatible with the digital input of the timer. For the X-ray tests, a 20 m long coaxial cable, with 50 Ω termination, was used to provide the sync signal (generated by the Clinac iX console outside the bunker where the LINAC was installed) to the input of the gated integrator, placed inside the bunker to avoid any detrimental effect on the detector response time. All instruments were remotely interfaced via a 20 m long LAN cable and controlled under a specifically developed Labview^®^ code. Before pulse-per-pulse measurements, the electronics were calibrated with a Keithley 6221 current source (Keithley Instruments & Products, Cleveland, OH, USA). As a result, a value of 88.49 pF was evaluated for the integration capacitance (*C_INT_*), which was then used to calculate the charge collected in correspondence of each X-ray pulse, as described in the following section.

## 3. Experimental Results

Figure 4 shows an example of a single-pulse measurement with an integration time set to 30 µs. Integration starts (S2 high) 2 µs after the sync signal rising edge; as expected, the integrator output signal starts to increase after a further 10 µs, generating a ramp for about 4 µs, then it flattens out. A 20 µs hold period (S1 high) completes the acquisition, and then a reset occurs (S1 and S2 low) before a new pulse arrives.

Figure 5 reports charge-per-pulse over time, measured for a total acquisition time of 4 s, at the two dose rates 1 Gy/min and 6 Gy/min, i.e., the minimum and the maximum DR values obtainable with our LINAC, respectively. Charge was calculated by multiplying the integrator output voltage by the capacitance *C_INT_* = 88.49 pF. As can be seen from the normal density functions fitted to the experimental histograms reported on the right, a peak centroid is located in both cases at about 85 pC, as expected from the results of the dosimeter preliminary characterization. In addition, by dividing the collected charge by the dosimeter sensitivity (302.2 nC/Gy), a dose-per-pulse value of about 3 × 10^−4^ Gy is obtained.

Table 1 summarizes the results (average <*Q*>, minimum *Q_min_* and maximum *Q_max_* charge) obtained on about 1000 pulses acquired at different DR values in the 1–6 Gy/min range. The average value of the charge-per-pulse measured by our system at all DRs was around <*Q*> ~84.68 pC, in excellent agreement with <*Q_est_*> ~84.60 pC, i.e., the value estimated after the continuous-mode measurements performed with the Keithley 6517A electrometer.

The proposed system also allows for the calculation of the charge *Q*(*t*) accumulated in the dosimeter after *t* seconds of pulsed irradiation. In Figure 6, the collected charge values accumulated during 5 s of X-ray irradiation are reported (blue line). Dose rate was set to 3 Gy/min. White squares represent the values *Q*_6517_(*t*) acquired with a Keithley 6517A electrometer in the same irradiation conditions. It is worth noting the excellent superposition of the data. In particular, the values of *Q*(*t*) and *Q*_6517_(*t*), measured after each second for the total acquisition time of 5 s, are summarized in Table 2. As can be inferred, *Q*(*t*) and *Q*_6517_(*t*) match within only 0.5%, thus validating the results of our prototype. However, it is worth stressing here that the time resolution of the proposed system (1 μs) greatly exceeds that of the 6517A electrometer (200 ms in the nC range [27]), thus enabling a real-time pulse-by-pulse processing even at dose rates of several kHz. It is also worth noting that, without any particular shielding of the implemented electronics, a noise amplitude lower than 1 mV_rms_ was evaluated for the acquired output signal, indicating a signal-to-noise ratio (SNR) higher than 60 dB.

## 4. Concluding Remarks

A complete measurement system based on a fast and high-quality single-crystal diamond dosimeter, coupled to specifically designed front-end readout electronics, was successfully tested under a medical LINAC (Varian, Clinac iX). It is worth observing that, by intrinsically implementing a sample-and-hold operation on the analog signal acquired from the detector, the front-end electronic is made simple, consisting of only an additional amplification/attenuation stage to fit the analog-to-digital converter input dynamics. This simplified design, along with the short response time of the diamond detector (in the ns range), allowed the system to be fast enough to measure, with a SNR > 60 dB, the dose delivered by every single X-ray pulse generated by the LINAC apparatus at a pulse frequency up to 360 Hz in the 1–6 Gy/min dose-rate range. Accurate pulse-by-pulse processing was therefore enabled, fully meeting the demand of modern radiotherapy techniques for reliable measurement systems ensuring real-time dose-per-pulse monitoring at a high dose rate. This current work represents progress from two perspectives: (1) it assesses the stability of a diamond detector over time, as well as the possible side effects caused by prolonged exposure to radiation; and (2) it upgrades proposed systems to realize a complete embedded system for data acquisition, processing and transfer.

## Figures and Tables

**Figure 1 materials-14-05203-f001:**
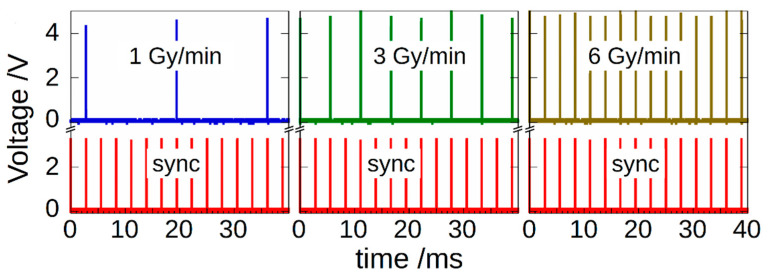
LINAC signal pulses at different dose rates (1, 3 and 6 Gy/min) synchronized with the 360 Hz sync signal (red peaks). Signal pulses are delayed for about 12 µs with respect to sync pulses rising edge [26]. Note that, in order to maintain the desired dose rate, the system periodically suppresses some pulses.

**Figure 2 materials-14-05203-f002:**
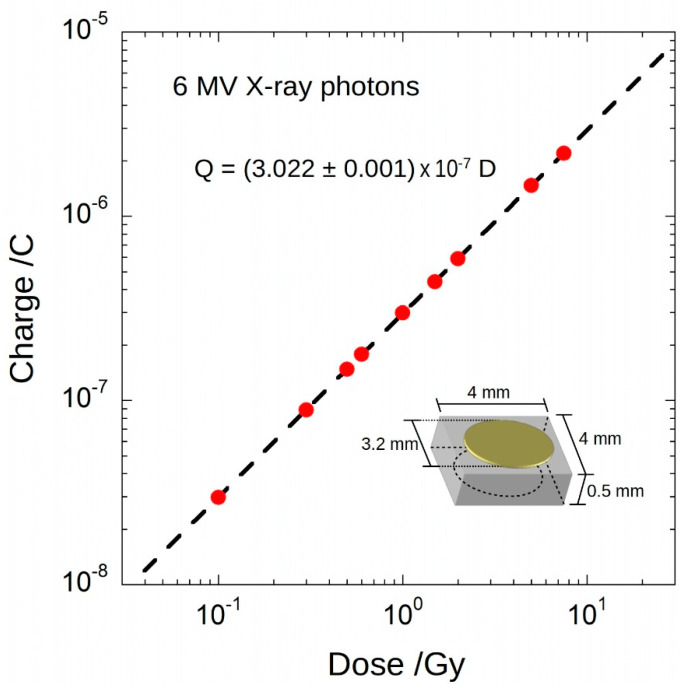
Charge as a function of dose acquired for 6₋MV photon beam. Error bars are smaller than symbols. Sensitivity is about 302 nC/Gy, with a nonlinearity lower than 0.5% in the investigated dose range. In the inset, the contact structure and the detector dimensions are shown.

**Figure 3 materials-14-05203-f003:**
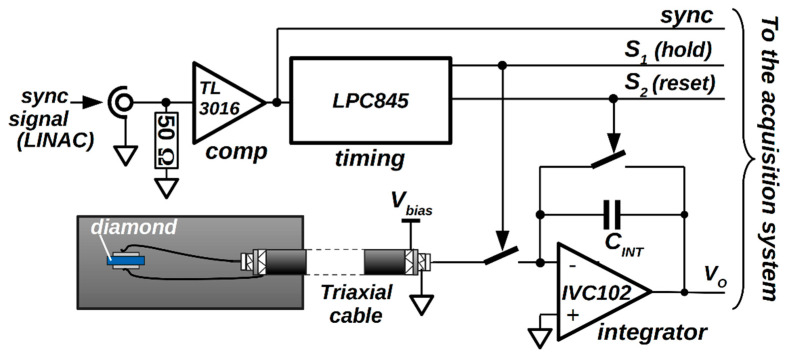
Block diagram of the implemented gated integrator for signal conditioning of the charge collected by the diamond detector.

**Figure 4 materials-14-05203-f004:**
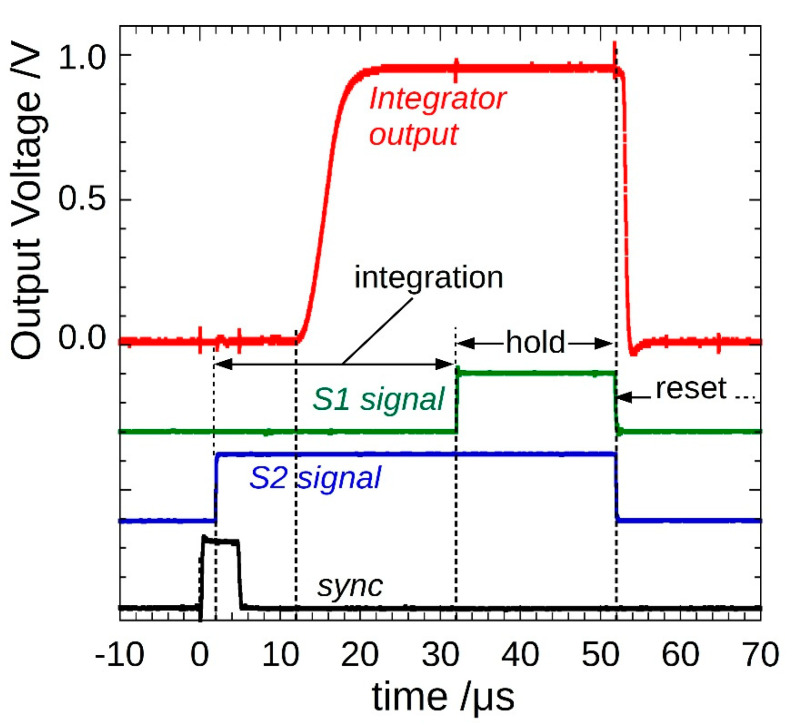
Typical gated₋integrator output voltage signal (red). Integration performed by IVC102, controlled by S1 and S2 signals (green and blue, respectively), started a few µs after the arrival of a pulse (sync signal, black). The system is ready for a new acquisition after 50 µs.

**Figure 5 materials-14-05203-f005:**
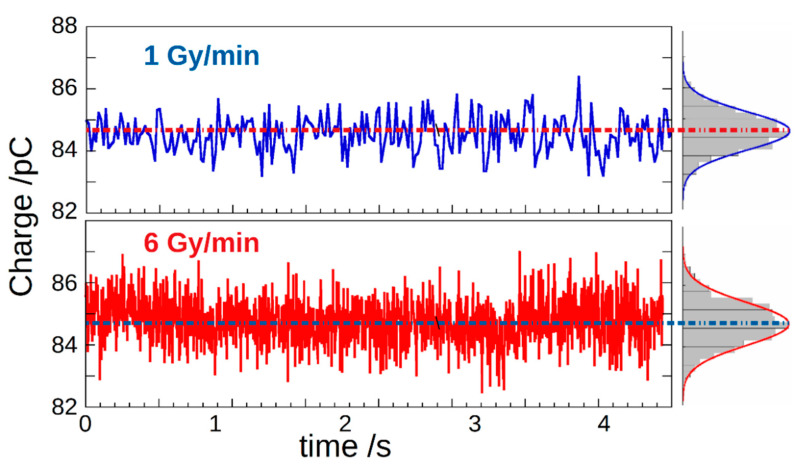
Charge-per-pulse values versus time measured at different DRs. Histograms on the right show their amplitude distribution.

**Figure 6 materials-14-05203-f006:**
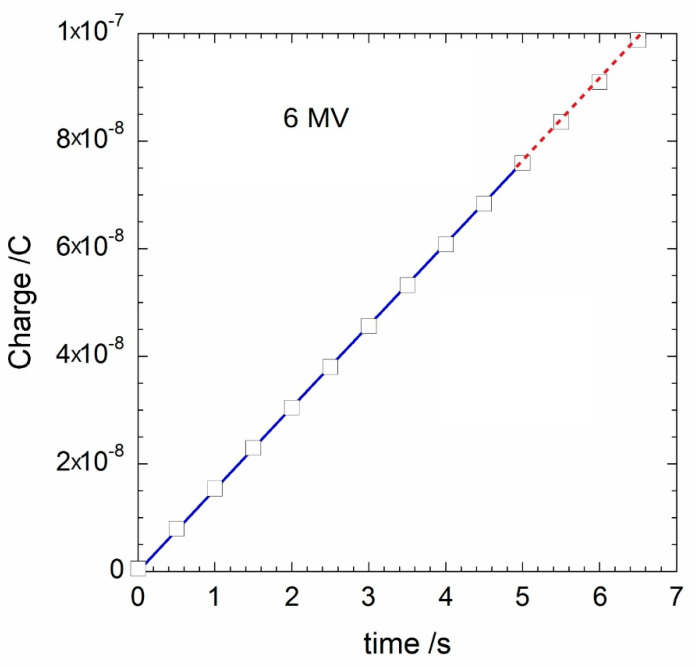
Accumulated charge versus time measured at 3 Gy/min for 6₋MV X-ray photons. Blue line represents the result acquired with the proposed readout electronics with the proposed readout electronics, whereas white squares refer to values acquired with a Keithley 6517A electrometer. Linear slope values were estimated to be (15.2490 ± 0.0003) nA and (15.209 ± 0.016) nA for our system and the Keithley 6517A, respectively. The difference between the estimated slope values is therefore lower than 0.3%, highlighting the perfect agreement between the two measuring systems.

**Table 1 materials-14-05203-t001:** Collected charge-per-pulse at different dose rates.

Dose Rate (Gy/min)	1	2	3	4	5	6
<*Q*> (pC)	84.60	84.63	84.67	84.60	84.46	84.78
*Q_max_* (pC)	88.83	82.01	82.64	81.13	81.97	82.48
*Q_min_* (pC)	86.99	86.68	86.93	86.76	87.01	87.00

**Table 2 materials-14-05203-t002:** Accumulated charge over time.

t (s)	1	2	3	4	5
*Q*(*t*) (nC)	15.28	30.54	45.75	60.98	76.10
*Q*_6517_(*t*) (nC)	15.34	30.51	45.70	60.90	76.08

## Data Availability

Study did not report any data.

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
