# Peer review of "A Diamond-Based Dose-per-Pulse X-ray Detector for Radiation Therapy"

_materials, 2021, doi:10.3390/ma14185203_

Round 1

Reviewer 1 Report

This contribution reports on the possibility to use diamond semiconductor detector for application of dosimetry in radiation therapy, with results obtained with prototype in medical accelerator.

The article is in excellent shape, well informative and presented. It can be published in present form, but I would suggest authors consider just one point:

- Diamond detectors are already used for very high rate beam monitoring in high energy physics experiments, for example The ATLAS Diamond Beam Monitor used in CERN from 2015 and operating at 300 kHz rate. It is not clear to this reviewer why such relevant technology from HEP was not included in the background.

Author Response

This contribution reports on the possibility to use diamond semiconductor detector for application of dosimetry in radiation therapy, with results obtained with prototype in medical accelerator.

The article is in excellent shape, well informative and presented. It can be published in present form, but I would suggest authors consider just one point:

- Diamond detectors are already used for very high rate beam monitoring in high energy physics experiments, for example The ATLAS Diamond Beam Monitor used in CERN from 2015 and operating at 300 kHz rate. It is not clear to this reviewer why such relevant technology from HEP was not included in the background.

R: We thank very much the reviewer for the comments. About ATLAS-DBM, the reviewer is completely right and we regret this omission. So, in the introduction, we added a paragraph mentioning ATLAS experiment and a related reference.

For instance, diamond detectors with long-term durability have already been successfully tested in high-radiation environments, such as those reproduced in ATLAS, a general-purpose particle physics experiment performed at the Large Hadron Collider (LHC) at CERN [25]”.

Reviewer 2 Report

This is a very well written manuscript. The scientific concepts are well thought out and executed. I do not see any need for major revisions.

It might serve the readers well to say what makes the CVD diamond that the authors chose apt for this study (as we know natural diamond is an insulator. so what makes the "optical grade" diamond suitable for this study?)

Does the contact pad or the diamond crystal's performance change over time? Do the authors notice any change in their readings, say after a million cycles of charge accumulation and dissipation.

Additional suggestions:

I believe there is novelty in this manuscript warranting publication. The use of diamond material as the core element of x-ray detector is a very nice approach. Of course, diamond has found many uses in areas as particle detectors at CERN, and in x-ray optics.
  The authors highlighted the versatility of diamond material in their introduction. However, they should elaborate some more on the properties of the particular diamond element they used in their experiments. What makes the "optical grade" diamond apt for this study? They should elaborate the material properties of this optical grade diamond and perhaps compare it with natural diamond, so the readers have some context for comparison.  

The radiation hardness of diamond was mentioned in introduction and in subsequent materials & methods section. While the dosage of LINAC pulses is nowhere near energetic enough to cause damage to diamond materials, I want the authors to make a comment on how repeated exposure of pulses affects their overall assembly. Does the PMMA cylinder housing offer sufficient protection? Specifically, the authors should mention what if any performance degradation occurs in the ultrathin contact pads they fabricated on the diamond material.

 I am satisfied with the sound scientific principles applied in constructing the readout electronics. I think the care taken in synchronizing the signals with LINAC pulses is self-evident in data shown.

Author Response

This is a very well written manuscript. The scientific concepts are well thought out and executed. I do not see any need for major revisions.

It might serve the readers well to say what makes the CVD diamond that the authors chose apt for this study (as we know natural diamond is an insulator. So what makes the "optical grade" diamond suitable for this study?)

R: First of all, thank you for the comments.

The development of the CVD technique has made it possible to obtain diamond samples of high quality and above all with specific controlled and reproducible properties at a relatively low cost. The fundamental aspect is to be able to control the concentration of impurities as nitrogen or boron. In the best quality natural diamond (type IIa) impurity concentration is not detectable by IR spectroscopy, and it is nominally of the order of few ppm. Conversely, high quality CVD-diamond (“electronic-grade” and “optical-grade”), the concentration of nitrogen and boron (<5 ppb and <0.5 ppb, as detected by means of EPR and SIMS techniques) is three orders of magnitude lower than that of type-IIa natural diamond. Sample grade also defines the cost and, in our case, optical-grade CVD-diamond represented a good choice for detector fabrication. It must be said that the CVD technique allows the growth of larger samples with uniformity characteristics far superior to those found with natural diamond samples. In the introductory section, few lines have been added according to the reviewer’s suggestion, aimed at providing more information to the reader about CVD diamond:

Chemical Vapor Deposition (CVD) represents the most widely used technique for the production of synthetic diamond, allowing for the growth of high quality poly- and mono-crystalline samples. In the case of Micro-Wave plasma-assisted CVD (MWCVD), which is the technique used for the diamond samples tested in this work, both the growth rate and the control of purity can be optimized: in “electronic-grade” and “optical-grade” samples, the concentrations of nitrogen and boron impurities can be reduced down to values lower than 5 and 0.5 ppb, respectively, i.e. about three orders of magnitude lower than those reported for the best (namely, type IIa) natural diamond [10]."

[10]: https://e6-prd-cdn-1.azureedge.net/mediacontainer/medialibraries/element6/documents/brochures/element_six_diamond_handbook_august_2020.pdf?ext=.pdf

Does the contact pad or the diamond crystal's performance change over time? Do the authors notice any change in their readings, say after a million cycles of charge accumulation and dissipation.

R: The authors thank the reviewer for the question. During the experiments, in which samples have been irradiated with several Gy, we did not observe neither contact pads degradation nor drift in data reading. Unfortunately, a dedicated investigation in this regard has not been conducted at this time. But we definitely agree with the reviewer that the assessment of the detector durability is crucial. For this purpose, we are currently performing new tests aimed also at verifying the stability of the detector over time. In the revised version of the manuscript, few lines have been added in the concluding remarks mentioning this:

Work is in progress in a two-fold way, in order to: 1) assess the stability over time of the diamond detector, as well as possible side effects caused by a prolonged exposure to radiation; 2) upgrade the proposed system, aimed at realizing a complete embedded system for data acquisition, processing, and transfer.”

Additional suggestions

I believe there is novelty in this manuscript warranting publication. The use of diamond material as the core element of x-ray detector is a very nice approach. Of course, diamond has found many uses in areas as particle detectors at CERN, and in x-ray optics.

R: The reviewer is right. in the introduction, we added a paragraph mentioning ATLAS experiment of CERN and a related reference.

For instance, diamond detectors with long-term durability have already been successfully tested in high-radiation environments, such as those reproduced in ATLAS, a general-purpose particle physics experiment performed at the Large Hadron Collider (LHC) at CERN [25]”.

The authors highlighted the versatility of diamond material in their introduction. However, they should elaborate some more on the properties of the particular diamond element they used in their experiments. What makes the "optical grade" diamond apt for this study? They should elaborate the material properties of this optical grade diamond and perhaps compare it with natural diamond, so the readers have some context for
comparison.

R: First of all, thank you for the comments.

The development of the CVD technique has made it possible to obtain diamond samples of high quality and above all with specific controlled and reproducible properties at a relatively low cost. The fundamental aspect is to be able to control the concentration of impurities as nitrogen or boron. In the best quality natural diamond (type IIa) impurity concentration is not detectable by IR spectroscopy, and it is nominally of the order of few ppm. Conversely, high quality CVD-diamond (“electronic-grade” and “optical-grade”), the concentration of nitrogen and boron (<5 ppb and <0.5 ppb, as detected by means of EPR and SIMS techniques) is three orders of magnitude lower than that of type-IIa natural diamond. Sample grade also defines the cost and, in our case, optical-grade CVD-diamond represented a good choice for detector fabrication. It must be said that the CVD technique allows the growth of larger samples with uniformity characteristics far superior to those found with natural diamond samples. In the introductory section, few lines have been added according to the reviewer’s suggestion, aimed at providing more information to the reader about CVD diamond:

Chemical Vapor Deposition (CVD) represents the most widely used technique for the production of synthetic diamond, allowing for the growth of high quality poly- and mono-crystalline samples. In the case of Micro-Wave plasma-assisted CVD (MWCVD), which is the technique used for the diamond samples tested in this work, both the growth rate and the control of purity can be optimized: in “electronic-grade” and “optical-grade” samples, the concentrations of nitrogen and boron impurities can be reduced down to values lower than 5 and 0.5 ppb, respectively, i.e. about three orders of magnitude lower than those reported for the best (namely, type IIa) natural diamond [10]."

[10]: https://e6-prd-cdn-01.azureedge.net/mediacontainer/medialibraries/element6/documents/brochures/element_six_diamond_handbook_august_2020.pdf?ext=.pdf

The radiation hardness of diamond was mentioned in introduction and in subsequent materials & methods section. While the dosage of LINAC pulses is nowhere near energetic enough to cause damage to diamond materials, I want the authors to make a comment on how repeated exposure of pulses affects their overall assembly. Does the PMMA cylinder housing offer sufficient protection? Specifically, the authors should mention what if any performance degradation occurs in the ultrathin contact pads they fabricated on the diamond material.

R: As commonly adopted for commercial dosimeters, the PMMA housing was chosen for its water-equivalence and well-known radiation hardness; indeed, as stated in a previous comment, we did not observe neither contact pads degradation nor drift in data reading during the experiments (in which samples have been irradiated with several Gy), indicating a good stability over time of the detector. Of course, we acknowledge the importance of further investigating on this aspect, as highlighted in the conclusions: “Work is in progress in a two-fold way, in order to: 1) assess the stability over time of the diamond detector, as well as possible side effects caused by a prolonged exposure to radiation; 2) upgrade the proposed system, aimed at realizing a complete embedded system for data acquisition, processing, and transfer.”

I am satisfied with the sound scientific principles applied in constructing the readout electronics. I think the care taken in synchronizing the signals with LINAC pulses is self-evident in data shown.

R: We thank very much the reviewer for the comment.

Reviewer 3 Report

The manuscript is clear and readable despite a general revision of the English language and typing errors are required. 

In the introduction I would like to read a few sentences that refer to medical treatments and cite references. Is this system able to detect dose X ray also when the radiotherapy is applied to  lung tumors? could the movement due to breathing cause false dose counts?

Author Response

The manuscript is clear and readable despite a general revision of the English language and typing errors are required.

R: We thank the reviewer for the comment. We have revised the manuscript and typos/grammar errors have been corrected.

In the introduction I would like to read a few sentences that refer to medical treatments and cite references.

R: We thank the reviewer for the suggestion. In the introduction, IMRT and VMAT techniques have been described in more details by adding a few lines, as well as three references: IMRT and VMAT are highly conformal techniques that allow irradiating complex and irregularly shaped tumors close to delicate organs that could be damaged if the absorbed dose exceeds a certain tolerance level: specifically, the dose is released by LINAC in a very short time interval, and is appropriately modulated to be maximum on the tumor mass and minimum at the level of organs at risk [5-7]. This ensures an optimal efficacy of treatment, as well as a lower incidence of side effects”.

Is this system able to detect dose X ray also when the radiotherapy is applied to lung tumors? could the movement due to breathing cause false dose counts?

R: The proposed detection system is thought to be used in the preliminary design and calibration phase of RT-plan definitions for the treatment of solid tumors. Detector characterization was performed by applying a 10 V bias voltage. Conversely, in-vivo dosimetry, i.e. during patient exposure, requires zero-biased detectors (i.e. diodes). We plan to exploit such a method with new detectors based on rectifying contacts.

In the introductory section of revised manuscript, according to the reviewer’s comment, we stress that our system can be “specifically and fruitfully used in the development of optimized RT plans for the treatment of solid tumors.”

Reviewer 4 Report

Very interesting paper and impressive results. I would recommend a few stylistic improvements (e.g. citations at the end of a sentence, using third person passive throughout the text). There are also a few grammatical and typographical mistakes to be addressed. 

In terms of improving the presentation of the results. I would suggest to provide a detailed drawing or photograph of the device to guide the reader on the geometry and size. I would also like to see a quantitative measure of the agreement shown in figure 6 (e.g a chi-square test or similar).

I assume to detector to the a thin oblong. EBRT will rotate around the central axis of the patient, so the angle of incidence and hence the effective amount of material presented to the incoming radiation will change. How is that taken into account for this device?

Author Response

Very interesting paper and impressive results. I would recommend a few stylistic improvements (e.g. citations at the end of a sentence, using third person passive throughout the text). There are also a few grammatical and typographical mistakes to be addressed.

R: We are grateful to the reviewer for the comment. The text was improved according to his/her suggestion.

In terms of improving the presentation of the results. I would suggest to provide a detailed drawing or photograph of the device to guide the reader on the geometry and size.

R: Following reviewer’s suggestion, a sketch of the diamond detector has been added as inset of Fig. 2, where pad and sample dimensions are reported.

I would also like to see a quantitative measure of the agreement shown in figure 6 (e.g a chi-square test or similar).

R: According to a linear fit of the experimental data, slope values of (15.2490 ± 0.0003) nA and (15.209 ± 0.016) nA were estimated for measurements conducted with our system and the Keithley 6517A, respectively. The difference between the estimated slope values is therefore lower than 0.3%, highlighting the accuracy of the measurement system proposed in this work and the agreement with a commercial device.

I assume to detector to the a thin oblong. EBRT will rotate around the central axis of the patient, so the angle of incidence and hence the effective amount of material presented to the incoming radiation will change. How is that taken into account for this device?

R: The authors thank the reviewer very much for the question. Angular dependence is currently under investigation. Tests are in progress to understand how the dosimeter response varies as a function of the angle of incidence of the radiation. It is reasonable to expect a variation of a few percent by varying the incidence angle in the polar direction, as pointed out by several works carried out with samples similar to those used in our work (see Ciancaglioni et al., (2012), Dosimetric characterization of a synthetic single crystal diamond detector in clinical radiation therapy small photon beams, Medical physics39, 4493-4501).